# Effects of Wheat Bran Applied to Maternal Diet on the Intestinal Architecture and Immune Gene Expression in Suckling Piglets

**DOI:** 10.3390/ani10112051

**Published:** 2020-11-06

**Authors:** Julie Leblois, Yuping Zhang, José Wavreille, Julie Uerlings, Martine Schroyen, Ester Arévalo Sureda, Hélène Soyeurt, Frédéric Dehareng, Clément Grelet, Isabelle P. Oswald, Bing Li, Jérôme Bindelle, Hongfu Zhang, Nadia Everaert

**Affiliations:** 1Association Wallonne de l’Élevage Asbl (AWÉ), 5590 Ciney, Belgium; jleblois@awenet.be; 2Precision Livestock and Nutrition Laboratory, TERRA Teaching and Research Centre, Gembloux Agro-Bio Tech, University of Liège, 5030 Gembloux, Belgium; yuping.zhang@doct.uliege.be (Y.Z.); J.wavreille@cra.wallonie.be (J.W.); julie.uerlings@gmail.com (J.U.); martine.schroyen@uliege.be (M.S.); Ester.ArevaloSureda@uliege.be (E.A.S.); jerome.bindelle@uliege.be (J.B.); 3State Key Laboratory of Animal Nutrition, Institute of Animal Sciences, Chinese Academy of Agricultural Sciences, Beijing 100193, China; zhanghf6565@vip.sina.com; 4Laboratory of Statistics, Informatics and Modelling Applied to Bioengineering, TERRA Teaching and Research Centre, Gembloux Agro-Bio Tech, University of Liège, 5030 Gembloux, Belgium; hsoyeurt@uliege.be; 5Valorisation of Agricultural Products Department, Walloon Agricultural Research Centre, 5030 Gembloux, Belgium; f.dehareng@cra.wallonie.be (F.D.); c.grelet@cra.wallonie.be (C.G.); 6Toxalim Research Centre in Food Toxicology, University of Toulouse, INRA, ENVT, INP-Purpan, UPS, 31027 Toulouse, France; isabelle.oswald@inra.fr; 7New Hope Liuhe Co., Ltd., Qingdao 266000, China; libing9@newhope.cn

**Keywords:** fiber, intestinal health, maternal transfer, piglets, weaning

## Abstract

**Simple Summary:**

This research was committed to revealing the potential effects of the use of a high percentage of wheat bran (WB) in the sow’s diets on the offspring’s growth and health, by measuring the bodyweight gain, the morphology of the intestine, as well as the expression levels of immune-related genes in the mucosa of the ileum and colon. Results indicate that adding 25% of wheat bran to the sow’s gestation and 14% to the lactation diet can affect the intestinal architecture and the expression of some inflammation genes, to some extent, in the ileal mucosa in the progeny.

**Abstract:**

The strategy of improving the growth and health of piglets through maternal fiber diet intervention has attracted increasing attention. Therefore, 15 sows were conducted to a wheat bran (WB) group, in which the sows’ diets included 25% of WB in gestation and 14% in lactation, and a control (CON) group, in which the sows’ diets at all stages of reproduction did not contain WB. The results show that maternal high WB intervention seems not to have an impact on the growth of the offspring or the villus height of the duodenum, and the ratio of villi/crypts in the duodenum and jejunum were all higher in piglets born from WB sows, which may indicate that WB piglets had a larger absorption area and capacity for nutrients. The peroxisome proliferator-activated receptor gamma (*PPARγ*) and interleukin 6 (*IL6*) expression levels were notably upregulated in the ileal mucosa of WB piglets, while no immune-related genes in the colonic mucosa were affected by the maternal WB supplementation. In conclusion, adding a high proportion of wheat bran to the sow’s gestation and lactation diet can affect the intestinal architecture and the expression of some inflammation genes, to some extent, in the ileal mucosa in the progeny.

## 1. Introduction

Wheat bran (WB), a major byproduct of the wheat milling process, rich in dietary fiber (DF), especially non-starch polysaccharides (NSPs), has been abundantly used in livestock and poultry feeds. Although DF used to be recognized as an anti-nutritional factor due to its negative effect on the utilization of nutrients [1,2], it is currently receiving more attention thanks to its beneficial effects on the gut health of monogastric animals as it is fermented in the hindgut [3,4]. Studies show that the use of WB in chickens or pigs could improve animals’ growth and health through improving intestinal morphology [5], changing the gut microbiota composition [6,7,8] and the profile of short chain fatty acids (SCFA), especially for butyrate [9,10,11]. Changes in the SCFA profile and microbiota due to fiber ingestion are assumed to be responsible for the improvement of the intestinal barrier and immune system [7,8,12].

In addition to the use of WB in the diet of weaned piglets, another strategy is the addition of WB to the maternal diet, to attempt to positively affect the intestinal health of the progeny [13,14]. Our previous study showed that a high maternal WB diet could not only change the sows’ gut microbiota, but also changed a few genera of their offspring’s colonic microbiota at the moment of weaning [15]. Gut bacteria have been shown to modulate the maturation of the immune system in the early life of piglets [16]. Furthermore, it is possible that changes in the gut microbes of sows and piglets may affect the immune response of piglets [13,17]. In addition to the effect of the microbiota, milk components also exert a maturation effect on the gut and have been shown to be affected by the sow’s diet [18]. High levels of WB in the diet of sows increased interleukin 10 (IL10) levels in the colostrum, and affected the piglet’s barrier function genes, shown by an upregulation of IL10 and zonula occludens-1 (ZO-1) and a downregulation of tumor necrosis factor alpha (TNF-α) [19]. 

In this study, we investigated the bodyweight, the villi height and crypt depth of the small intestine, and the expression of gut barrier and inflammatory response related genes in the ileal and colonic mucosa of piglets at weaning (four weeks of age) by high-throughput qPCR, in order to reveal the effects of high amounts of WB in a sow’s diet on offspring’s growth and health.

## 2. Materials and Methods 

### 2.1. Ethical Approval

The animal experiment was approved by the Ethical Committee of the University of Liège (protocol number 1661). 

### 2.2. Animals and Experimental Design

Fifteen sows were divided into two groups, in which 7 sows (3 primiparous sows and 4 multiparous) were in the wheat bran (WB) group, and 8 of them (4 primiparous and 4 multiparous) were in the control (CON) group. From the 43rd day of gestation until weaning, the WB sows were fed with a WB enriched diet, which contained 25% of WB in gestation and 14% of WB in lactation. Sows in the CON group were fed a diet devoid of WB during the same period. Both the diets were formulated to be iso-energetic and iso-nitrogenous and to meet the nutritional requirements of the sows according to the National Research Council (NRC) recommendations (Appendix A). The piglets in each group were weighed weekly and weaned after 4 weeks of suckling. For a detailed description of the housing system and the feed, we refer to Leblois et al. [15].

Fifteen ml of colostrum was collected manually from functional tits within the first 3 h after the onset of farrowing, and 15 mL of milk was sampled weekly after intramuscular injection of 1.5 mL of oxytocin 10 Un/mL (V.M.D., Arendonk, Belgium). The concentration of fat, protein and lactose was determined in the colostrum and milk samples by Fourier transform infrared spectroscopy on a Standard Lactoscope FT-MIR automatic (Delta Instruments, Drachten, The Netherlands)—the same methods and equipment are used as described by Leblois et al. [20]. Capture enzyme-linked immunosorbent assays (ELISAs) were used to quantify the total concentrations of immunoglobulin (Immunoglobulin A (IgA), IgM, and IgGin colostrum and milk (Bethyl Laboratories, Montgomery, AL, USA and R&D Systems, Oxon, UK), following the manufacturer’s recommendations (Leblois et al., 2018).

At 4 weeks of age, 16 female piglets with a similar bodyweight (BW) as the mean litter weight (CON: 7.6 ± 0.2 kg vs. WB: 7.8 ± 0.2 kg) were euthanized (8 piglets/treatment, 2 piglets/sow). The drug used for anesthesia was an injection of a mix of xylazine/Zoletil 100 (4 mg of xylazine, 2 mg of zolazepam and 2 mg of tilamine/kg BW) (Kela SA, Hoogstraten, Belgium; Virbac, Leuven, Belgium), followed by a T61 injection (0.1 mL/kg BW) for euthanasia (Intervet Belgium N.V., Brussel, Belgium). The ileal and colonic mucosa were snap frozen in liquid nitrogen and then transferred to storage in a −80 °C freezer until further RNA extraction. Five cm of tissue samples from the duodenum, jejunum and terminal ileum were collected and dehydrated in 4% formaldehyde for 48 h followed by storage in an ethanol 70% solution. Histomorphological analyses were completed as described by Leblois et al. (2018).

### 2.3. qPCR

Total RNA of ileal and colonic mucosa was extracted as described by Uerlings [21]. Quantitative PCR (qPCR) was performed in a BioMark HD system, using a 48.48 Dynamic Array (Fluidigm Corporation), which combines 48 samples with 48 primer sets. These included 10 housekeeping genes, 4 of them were selected for the relative quantification. These four housekeeping genes were, for ileum, the genes encoding for TATA box binding protein (*TBP*), peptidylprolyl isomerase A (*PPLA*), hypoxanthine phosphoribosyl transferase 1 (*HPRT1*), β-actin (*ACTB*) and, for colon, TBP, glyceraldehyde-3-phosphate dehydrogenase (*GADPH*), ribosomal protein L4 (*RPL4*), *RPL13α, PPL32* and tyrosine 3-monooxygenase/tryptophan 5-monooxygenase activation protein zeta (*YWHAZ*). Nine inflammation pathway genes were tested, including serine/threonine-protein kinase (*AKT1*), mitogen-activated protein kinase 14 (*MAPK14*), myeloid differentiation primary response 88 (*MyD88*), nuclear factor kappa B1 (*NFκB1*), nuclear factor-kappa B inhibitor alpha (*NFκBIα*), nucleotide-binding oligomerization domain-containing protein 1 (*NOD1*), peroxisome proliferator-activated receptor gamma (*PPARγ*), toll-like receptor 2 (*TLR2*) and toll-like receptor 4 (*TLR4*). In addition, 13 pro-inflammatory genes, such as those encoding for chemokine ligand 5 (*CCL5*), cyclooxygenase 2 (*COX2*), C-X-C motif chemokine 10 (*CXCL10*), defensin beta-1 (*DEFβ1*), defensin beta-4A (*DEFβ4A*), interferon beta (*IFNβ*), interleukin 1 beta (*IL1β*), IL18, IL6, IL8, interleukin-1 receptor antagonist (*ILRN1*), monocyte chemoattractant protein 1 (*MCP1*) and tumor necrosis factor alpha (*TNFα*), were also determined. To test intestinal barrier integrity, 12 genes encoding for the following proteins were examined: caspase 3 (*CASP3*), E-cadherin (*CDH1*), claudin-1, claudin-3, claudin-4, epidermal growth factor receptor (*EGFR*), tricellulin (*MARVELD2*), mucin 1 (*MUC1*), occludin, transforming growth factor beta 1 (*TGFβ1*), villin 1 (*VIL1*) and zonula occludens-1 (ZO-1). The standard curve method was used to compare the samples, comparing all Cycle threshold (Ct) values relatively to a standard curve. The geometrical mean of the four most stable housekeeping genes was used for normalization. Primers can be found in Appendix A.

### 2.4. Statistical Analyses

All statistical analyses were performed using the Statistical Analysis System (SAS) 9.2 software (Cary, NC, USA). Piglet bodyweight was analyzed with the MIXED repeated procedure of SAS, using time as repeated factor, sow as random factor and sex, and parity and maternal diet as fixed factors. Milk composition was analyzed with the repeated MIXED procedure of SAS, including time (week) as repeated effect, sow as random effect and diet and parity as fixed factor. For the high-throughput qPCR, results were analyzed with the one-way ANOVA considering the maternal treatment as fixed factor. All averages are shown in mean ± SEM. Next, *q*-values were obtained using a false discovery rate correction with the linear method of Benjamini and Hochberg. Significance was obtained with *q* < 0.05.

## 3. Results

### 3.1. Zootechnical Results

The piglets’ BW was not affected by the maternal WB diet (*p* > 0.05); however, an effect of sex was observed (*p* = 0.017), the male piglets being heavier than females at every time point (Appendix A).

### 3.2. Milk Composition

As shown in Table 1, most of the milk components had significant changes over time, especially when comparing colostrum with regular milk. The contents of protein and immunoglobulin were observed to drop sharply after one week in both treatments, while the concentration of fat and lactose increased (*p* < 0.001). Protein and fat concentration in colostrum and milk were not influenced by maternal treatments but influenced by the parity of the sows (*p* = 0.04 and *p* = 0.03, respectively). However, the lactose percentage was not affected by the parity of the sows but globally higher in the WB group than in the CON group (*p* = 0.046), both on week two and week three. There were no effects on the measured immunoglobulins by maternal diet treatments. In addition, a parity effect was observed in sows’ colostrum, the primiparous sows contained lower amounts of IgA than multiparous sows in colostrum (11.97 ± 0.43 mg/mL and 15.03 ± 1.31 mg/mL, respectively).

### 3.3. Histomorphology Analysis of the Small Intestine

The villi height and crypt depth were used to represent the morphology of the small intestine (Figure 1). Our results show that the villi height in the duodenum of piglets was significantly higher in the WB group than that in the CON group (*p* < 0.05), which is also the reason for the increase in villi/crypt (V/C) ratio (*p* < 0.01). The crypts depth in the jejunum tended to be lower for the WB piglets than those of the CON piglets (*p* < 0.1), thereby the villi/crypt ratio was significantly higher for the WB piglets (*p* < 0.05). However, the villi height, the crypt depth and the (V/C) ratio parameters were found to be unaffected by the maternal treatments for the ileum of piglets (*p* > 0.05).

### 3.4. Relative Gene Expression in Ileum and Colon Mucosa

In the ileal mucosa, the expression of PPARγ, a gene in the inflammation signaling pathway, and IL6, a pro-inflammatory gene, were significantly higher when looking at the *p*-value (mean ± SEM of 1.02 ± 0.12 for WB vs. 0.61 ± 0.15 for CON, *p* = 0.049, *q* = 0.813, for *PPARγ*; mean ± SEM of 1.37 ± 0.12 for WB vs. 1.03 ± 0.096 for CON, *p* = 0.047; *q* = 0.813 for *IL6*) in the piglets born from WB sows compared to piglets born from control sows. No other gene in the ileum differed between treatments and neither did the genes determined in the colon (Appendix A). 

## 4. Discussion

The results from the present study show that WB supplementation in the diet of sows during gestation and lactation did not markedly affect the growth performance of their offspring. This result is in agreement with several previous studies [22,23,24], showing that the maternal high fiber diet has no or limited effects on the performance of the progeny. 

The villus height to crypt depth ratio is a useful criterion for estimating the likely digestive or absorptive capacity of the small intestine [25]. Li et al. reported an increased crypt depth of the jejunum in weaned piglets with an increased maternal insoluble fiber/soluble fiber ratio [26]. Similarly, in the present study, the villi/crypt ratio of duodenum and jejunum were significantly higher in piglets around four weeks. The question remains if the higher lactose concentration in the WB sow’s milk has played a role on the intestinal architecture of the piglets, and if this phenomenon would have given the piglets an improved condition for the stressful weaning event, or whether it would have impacted the performance long term. 

On the other hand, the equal amounts of protein, fat, and immunoglobulin content in the milk of the two groups did not present a more favorable situation for the piglets, as they received the same passive immunity.

In our previous study, we showed that the microbiota of the colon of these piglets was affected to a certain extent at weaning. Three genera significantly differed between treatments, and six genera tended to differ [15]. Although the exact mechanisms are not always known, shifts in gut microbiota composition and density can alter immune homeostasis and protective responses in the gut. Hence, we focused on some important genes involved in the immune response and investigated if feeding WB would influence their expression. Proinflammatory cytokines, such as *TNFα*, trigger inflammatory response and have negative effects on intestinal integrity and epithelial function [27,28]. In contrast, anti-inflammatory cytokines, such as *PPARγ* and *IL10*, are a series of immunoregulatory molecules that control the proinflammatory cytokine response [29,30,31,32]. It has been reported that some cytokines, such as *IL6*, have opposite properties related to their form being soluble or transmembrane, resulting in either pro- or anti-inflammatory working mechanisms, respectively [33,34]. The present study shows no differences in the mRNA expression of the colonic mucosa between the piglets from the two examined groups. However, *IL6* and *PPARγ* were upregulated in the piglets’ ileum of the WB group. Interestingly, the upregulation of *IL6* can play an important role in mucosal protection, associated with reducing infection-induced apoptosis in the epithelium and subsequent ulcerations [35]. We speculate therefore that the function of IL6 in the piglet’s ileal mucosa in this study might be anti-inflammatory and plays a key role in barrier integrity. It remains to be investigated whether piglets, being the offspring of sows receiving wheat bran during late gestation, would cope better with weaning stress 

## 5. Conclusions

The maternal high WB supplementation (25% in gestation and 14% in lactation) did not impact piglets’ growth from birth until weaning. Nevertheless, at four weeks of age, piglets from the WB group had a higher villi height in the duodenum, and villi/crypt ratio in duodenum and jejunum. Furthermore, the increased ileal *IL6* and *PPARγ* mRNA levels of piglets born from WB sows warrant further investigation.

## Figures and Tables

**Figure 1 animals-10-02051-f001:**
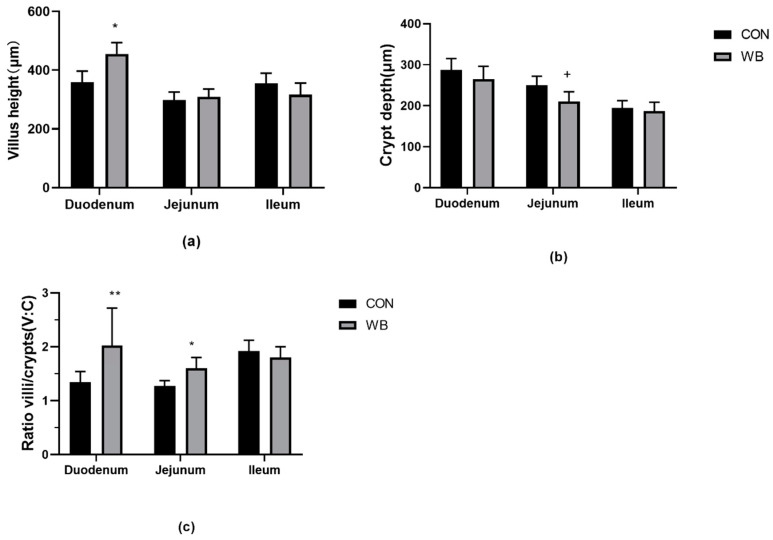
The villus height (**a**), crypt depth (**b**) and villi/crypt (V:C) ratio (**c**) of piglets born from CON (black bar) or WB (grey bars) sows, *n* = 8, values are shown as mean ± SEM, * means *p* < 0.05, ** represents *p* < 0.01 + represents *p* < 0.10, the unit of height and depth is μm.

**Table 1 animals-10-02051-t001:** Fat, protein, lactose percentage and IgA, IgG and IgM concentrations in colostrum and milk samples collected on a weekly basis after farrowing from control (CON) or wheat bran (WB) sows. Values are presented as mean and SEM given for each period, *n* = 3 or 4 for each parity within a treatment; W1, W2 and W3 represents week 1, week 2, and week 3; P1 means the first parity of sows, and similarly the *p* ≥ 2 means the parity of sows are more than 2, which are multiparous sows. * points to interaction effects between the respective parameters.

Sample	Treatment	Parity	Protein (%)	Fat (%)	Lactose (%)	IgA (mg/mL)	IgG (mg/mL)	IgM (mg/mL)
Colostrum	CON	1	19.49	6.48	2.6	12.63	68.58	4.99
	CON	≥2	18.77	6.21	2.6	14.81	59.74	4.73
	WB	1	18.36	6.63	2.66	11.48	65.69	4.23
	WB	≥2	20.4	6.08	2.56	15.25	71.31	4.21
	Global SEM		0.3	0.14	0.03	0.81	3.46	0.32
Milk W1	CON	1	6.1	10.26	4.75	1.84	0.42	0.92
	CON	≥2	6.22	9.29	4.59	2.3	0.39	1.21
	WB	1	5.83	10.37	4.71	2.49	0.34	0.95
	WB	≥2	6.1	8.59	4.77	2.57	0.48	1.35
	Global SEM		0.1	0.51	0.05	0.16	0.05	0.11
Milk W2	CON	1	5.95	10.93	4.79	2.26	0.27	1.1
	CON	≥2	6.1	9.02	4.82	2.55	0.32	1.09
	WB	1	5.43	9.68	4.85	2.36	0.24	0.97
	WB	≥2	5.8	9.55	4.93	2.96	0.27	1.19
	Global SEM		0.1	0.28	0.02	0.17	0.02	0.07
Milk W3	CON	1	6.31	10.84	4.81	3.22	0.21	1.01
	CON	≥2	6.15	8.88	4.84	3.56	0.2	0.95
	WB	1	5.74	8.31	4.95	3.77	0.16	0.98
	WB	≥2	5.98	9.4	4.91	3.37	0.16	1.03
	Global SEM		0.08	0.38	0.03	0.23	0.02	0.08
Overall *p*-values								
Treatment			0.16	0.23	<0.05	0.77	0.52	0.58
Time			<0.001	<0.001	<0.001	<0.001	<0.001	<0.001
Parity			0.04	0.03	0.75	0.06	0.82	0.76
Treatment*Week			0.48	0.48	0.74	0.68	0.8	0.31
Treatment*Parity			<0.01	0.16	0.58	0.82	0.28	0.73
Time*Parity			0.4	0.63	0.2	0.03	0.34	0.19
Treatment*Time*Parity			0.12	0.06	0.25	0.04	0.28	0.9

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
