# Peer review of "Effects of Wheat Bran Applied to Maternal Diet on the Intestinal Architecture and Immune Gene Expression in Suckling Piglets"

_animals, 2020, doi:10.3390/ani10112051_

Round 1
Reviewer 1 Report
In this study, Leblois et al investigated the potential effects of high percentage of wheat bran in the gestating (25%) and nursing (14%) sow’s diets on piglet’s growth and health. They found that the high fiber diet did not affect body weight gain of piglets, but affected the intestinal architecture and increased expression levels of two of tested genes (IL-6 and PPARgamma) related to immune response in the ileal muscosa in the offspring. Their observations seem interesting and worthy of further study the long term effect on offspring growth and health. This manuscript is overall well organized and well written, but quite a few issues need fixing.
- Line 75, “avoid” or “devoid”?
- Lines 102-119 and elsewhere, if the authors meant genes, which should be italic; otherwise some rewording might be better, such as “genes encoding TATA boxing binding protein (TBP). Ribosomal protein L (RPL) is a big family of protein consisting of the large subunit of ribosome. Please specify the exact one.
- The authors didn’t describe the details about how qPCR data was analyzed. Did they used the 2^(-deltadeltaCt) method to quantify gene expression change? What values were used as dependent variable for one-way ANNOVA analysis? –deltadeltaCt? If yes, please describe how the primer amplification efficiency was determined. Do the primer sets have a ~100% amplification efficiency? If not, you might refer to https://www.ncbi.nlm.nih.gov/pmc/articles/PMC4280562/ to correct for amplification efficiency.
- The authors define statistical significance as test with q < 0.05 (Line 127) after correction for multiple testing using the BH method, but between lines 161 and 166, they used FDR, which is equivalent to q-value. Being consistent might be better. More importantly, the q values, or FDR, for PPARgamma and IL6 are 0.813, which are much greater than 0.05, but the authors considered their expression “were significantly higher (Line 162)”. Please explain.
- Line 175, “Yang Li et al” should be “Li et al”.
- Lines 195-201, the authors considered that IL-6 is a pro-inflammatory cytokine, they concluded that “these two genes (IL6 and PPARgamma) are giving contradictory results…”. However, in the literature (https://mdpi-res.com/d_attachment/ijms/ijms-20-06097/article_deploy/ijms-20-06097.pdf and https://www.sciencedirect.com/science/article/pii/S0167488911000425), IL-6 has two forms, transmembrane and soluble, which have two opposite immune properties: anti-inflammatory and pro-inflammatory, respectively. The authors might consider the two opposite roles of IL-6 and the context of mucosa.
- In the reference list, there is some inconsistency with the journal titles, in terms of upper case/lower case, and abbreviation/full title. Please refer to the journal’s requirements.
- By the way, could you discuss the cost effect of supplementing high percentage of WB in the sow’s diets, and feed efficiency effect?
Author Response
Response to Reviewer 1 Comments
Point 1:Line 75, “avoid” or “devoid”?
Response: This should be “devoid” indeed, we have changed according to your advice (see line 78).
Point 2:Lines 102-119 and elsewhere, if the authors meant genes, which should be italic; otherwise some rewording might be better, such as “genes encoding TATA boxing binding protein (TBP). Ribosomal protein L (RPL) is a big family of protein consisting of the large subunit of ribosome. Please specify the exact one.
Response: In the section 2.3 qPCR, we changed our wording as to ‘the genes encoding for’. In brackets we give the gene name, which we put in italic. Everywhere else, when we talk about the gene, we have put it in italic as well. In addition, we changed RPL in this section into the specific family members we analyzed, RPL4, RPL13a and RPL32 (see line 109).
Point 3:The authors didn’t describe the details about how qPCR data was analyzed. Did they used the 2^(-deltadeltaCt) method to quantify gene expression change? What values were used as dependent variable for one-way ANNOVA analysis? –deltadeltaCt? If yes, please describe how the primer amplification efficiency was determined. Do the primer sets have a ~100% amplification efficiency? If not, you might refer to https://www.ncbi.nlm.nih.gov/pmc/articles/PMC4280562/ to correct for amplification efficiency.
Response: No, we did not use the 2^(-deltadeltaCt) method, the method we used was the relative standard curve method. This means that on the Fluidigm, aside from samples, we added 6 dilutions of a pooled sample in duplicate and referred all the Ct values of our samples of interest to the standard curve of that pooled sample for all genes. This way the specific efficiency of each primer is directly taken into account into the calculation, and is thus corrected for. We added that we used the relative standard curve method more clearly in the manuscript (see lines 124-126).
Point 4:The authors define statistical significance as test with q < 0.05 (Line 127) after correction for multiple testing using the BH method, but between lines 161 and 166, they used FDR, which is equivalent to q-value. Being consistent might be better. More importantly, the q values, or FDR, for PPARgamma and IL6 are 0.813, which are much greater than 0.05, but the authors considered their expression “were significantly higher (Line 162)”. Please explain.
Response: We changed FDR to q value throughout the whole manuscript now, only indicating in the material and methods that it is indeed a false discovery rate correction. And yes, the q-value of all the genes was greater than 0.05, suggesting not many gene expression differences in colon and ileal mucosa between the WB and CON piglets, that’s why we look at the p-value. However, only the p-value of PPARgamma and IL6 were lower than 0.05. We added that the level of significance was looked at at the p-value level (see line 175).
Point 5:Line 175, “Yang Li et al” should be “Li et al”.
Response: We have changed it (line 193).
Point 6:Lines 195-201, the authors considered that IL-6 is a pro-inflammatory cytokine, they concluded that “these two genes (IL6 and PPARgamma) are giving contradictory results…”. However, in the literature (https://mdpi-res.com/d_attachment/ijms/ijms-2006097/article_deploy/ijms-20-06097.pdf and https://www.sciencedirect.com/science/article/pii/S0167488911000425), IL-6 has two forms, transmembrane and soluble, which have two opposite immune properties: anti-inflammatory and pro-inflammatory, respectively. The authors might consider the two opposite roles of IL-6 and the context of mucosa.
Response: We thank the reviewer for this suggestion, and added the references. We described the fact that IL6 has different functions with regard to its form (line 211-213). We also added another paper (https://pubmed.ncbi.nlm.nih.gov/18453602/) and speculated the function of IL6 in the ileal mucosa being anti-inflammatory and useful in mucosal integrity in our situation (line 218-221).
Point 7:In the reference list, there is some inconsistency with the journal titles, in terms of upper case/lower case, and abbreviation/full title. Please refer to the journal’s requirements.
Response: We have changed them according to the journal’s requirements.
Point 8:By the way, could you discuss the cost effect of supplementing high percentage of WB in the sow’s diets, and feed efficiency effect?
Response: We discussed this point with nutritionists. Usually, they add around 12.5% of wheat bran in the diet of gestating sows, and they use a lot of different ingredients (that are rich in fibers) to formulate the diets for sows according to their requirements and taking the fluctuating prices of the ingredients into account. Therefore, in practice, besides cereals, they never really add an ingredient in such high quantities as we tested in our experiment (25% in gestating diets). In this experiment, our purpose was to solve the question on the possible transgenerational effect of the high wheat bran diet on the performance, gut morphology and intestinal gene expression of the progeny. Therefore, we created the diets containing wheat bran in high amounts (as high as possible, without focusing on the economical side). So due to these different points, it is impossible to include a statement on the cost effect of these WB diets.
Reviewer 2 Report
The author intervened the sow’s diets through a high percentage of wheat bran (WB) to investigate the effects on growth and immunity of offspring.
- The experimental design is disorder and confusing. The whole manuscript does not sound scientific.
- Line 41: The first letter of keyword should not be capitalized.
- Line 58: Deleted the blank before “changed a few…”
- How to ensure the diet avoid of WB supplementation?
- Please provide the composition and nutrient levels of experimental diets table during gestation and lactation in the materials and methods to present the nutrient levels and whether meet the NRC (2012) recommended requirements. Or is this information available in supplement?
- Line 74: “The sow diets included 25% of WB in gestation”, Is 25% the best level or dosage? Please provide the reference for the level. And whether your result “WB supplementation did not markedly affect their offspring’ growth performance” is relevant to the WB level?
- Line 71: Why did you choose 15 sows and divide 7 sows (3 primiparous sows and 4 multiparous) into the WB group, and 8 of them (4 primiparous and 4 multiparous) into the control (CON) group? Why not 8 vs. 8?
- Line 94: Please divide the materials and methods section into such as “Ethical Approval”, “Animals and Experiment Design”, “q-PCR”, and “Statistical Analysis”.
- Please add an “Ethical Approval” into the materials and methods not a sentence in line 70.
- Please add the data in line 131 and Line 161. Why the data not shown?
- Add a comma behind “IgG” in the title of Table 1 and check the problem in the manuscript such as “week 2” in line 146 and line 157. The format of Table 1 is disordered, correct it.
- Line 148: Please add the morphological structure (H.E.) analysis of small intestine into the result of the histomorphology.
- The all p value should be italicized. Please check.
- Line 94: “he contents of protein and immunoglobulin were observed to drop sharply” which group dropped and increased?
- Line 154: Check the font format of “none of”.
- Line 88: “see also Leblois et al., 2018” change to “Leblois et al., 2018”.
- Line 173: the explanation for higher lactose is that “WB prioritized nutrients for the offspring rather than to the sow” was illogical. Please add reference to explain.
- Line 187: “Three genera significantly differed between treatments, and 6 genera tended to differ”. Does this sentence have any logical relation to the below?
- Line 206: Since your result showed the maternal high WB supplementation did not impact piglets’ growth until weaning”, you should not mention the following “may affect growth on the long term”. The histomorphology result is paradoxical.
Author Response
Dear Editors and Reviewers:
We are grateful for the valuable comments and suggestions made by the reviewers on our manuscript entitled “Effects of wheat bran applied to maternal diet on the intestinal architecture and immune gene expression in suckling piglets”, which were very helpful for improving our paper. We have thoughtfully revised the manuscript and highlighted all changes by track changes.Please see the attachment.
Response to Reviewer 2 Comments
Point 1:The experimental design is disorder and confusing. The whole manuscript does not sound scientific.
Response: We have added supplementary tables and figures, according to your and other reviewers’ requirements., which would make the manuscript easier to understand. We have also explained some improper words (replace avoid with devoid) and confusing descriptions in the text (the specific questions and explanations, please refer to your point 17 and point 19).
Point 2:Line 41: The first letter of keyword should not be capitalized.
Response: We have changed according to your advice (line 41).
Point 3:Line 58: Deleted the blank before “changed a few…”
Response: We have changed according to your advice (line 58).
Point 4:How to ensure the diet avoid of WB supplementation?
Response: During the first 43 days of gestation, we made sure all the sows received a diet devoid of WB. Then we split up the sows into two groups and gave them the experimental diets, where one included the WB, while the control group did not have any wheat bran. We did this experiment in collaboration with a feed company, who helped in the formulation of the diet.
Point 5:Please provide the composition and nutrient levels of experimental diets table during gestation and lactation in the materials and methods to present the nutrient levels and whether meet the NRC (2012) recommended requirements. Or is this information available in supplement?
Response: We added a supplementary table in the material and methods part (Supplemental Table S1).
Point 6:Line 74: “The sow diets included 25% of WB in gestation”, Is 25% the best level or dosage? Please provide the reference for the level. And whether your result “WB supplementation did not markedly affect their offspring’ growth performance” is relevant to the WB level?
Response: To make sure that our experimental WB diet was iso-energetic and iso-nitrogenous, and followed the NRC requirements, we could include a max of 25% of WB in the gestation diet, and only 14% of WB in the lactation diet. Since we wanted to investigate any potential effect of WB, we tried this maximum amount possible.
Point7:Line 71: Why did you choose 15 sows and divide 7 sows (3 primiparous sows and 4 multiparous) into the WB group, and 8 of them (4 primiparous and 4 multiparous) into the control (CON) group? Why not 8 vs. 8?
Response: We first had 8 sows in each treatment, but as one sow needed an antibiotic treatment during gestation, we excluded her and her litter from the experiment. Therefore, we have 7 sows in the WB group.
Point 8:Line 94: Please divide the materials and methods section into such as “Ethical Approval”, “Animals and Experiment Design”, “q-PCR”, and “Statistical Analysis”.
Response: We have changed according to your advice.
Point 9:Please add an “Ethical Approval” into the materials and methods not a sentence in line 70.
Response: We have changed according to your advice.
Point 10:Please add the data in line 131 and Line 161. Why the data not shown?
Response: We have added the data according to your advice. Supplementary figure 1 shows the piglet’s bodyweight over time. This figure shows a clear difference in gender but no difference due to treatment over the whole period of 4 weeks before weaning. Supplemental table 3 and 4 show the qPCR data of ileum and colon respectively.
Point 11:Add a comma behind “IgG” in the title of Table 1 and check the problem in the manuscript such as “week 2” in line 146 and line 157. The format of Table 1 is disordered, correct it.
Response: We added the oxford comma wherever necessary as you suggested. Table 1 has been fixed as well.
Point 12:Line 148: Please add the morphological structure (H.E.) analysis of small intestine into the result of the histomorphology
Response: We are not really sure what the reviewer would like to see. The data are shown in a figure. We don’t think that a picture of the intestinal sections is necessary, as this is nowadays a parameter that is widely used, and it would not add any information on the morphological result.
Point 13:The all p value should be italicized. Please check.
Response: We have changed all of them in this manuscript according to your advice.
Point 14:Line 94(134): “he contents of protein and immunoglobulin were observed to drop sharply” which group dropped and increased?
Response: We have added “in both treatments” (line 143).
Point 15:Line 154: Check the font format of “none of”
Response: We have changed the format.
Point 16:Line 88: “see also Leblois et al., 2018” change to “Leblois et al., 2018”.
Response: We have deleted “see also”.
Point 17:Line 173: the explanation for higher lactose is that “WB prioritized nutrients for the offspring rather than to the sow” was illogical. Please add reference to explain
Response: we have deleted this statement. Our hypothesis was that the nutrients are mainly going to the milk instead of to the sow (so are for the progeny). But this is very hypothetical and we did not find literature to further support this statement. Yet, we want to draw the attention to the reader that there was a significant effect. Please see Line 179~180.
Point 18:Line 187: “Three genera significantly differed between treatments, and 6 genera tended to differ”. Does this sentence have any logical relation to the below?
Response: We added a logic connection between microbiota and immune response. We added that although the exact mechanisms are not always known, shifts in microbiota have been seen to alter immune response in the gut and since we did previously find changes in microbiota in our piglets, we here investigated immune genes using qPCR. The research question here was if, since WB significantly alters the presence of some genera in the piglets, the expression of these genes would change.
Point 19:Line 206: Since your result showed the maternal high WB supplementation did not impact piglets’ growth until weaning”, you should not mention the following “may affect growth on the long term”. The histomorphology result is paradoxical.
Response: we have deleted the description “may affect growth on the long term”. We also started the sentence with Nevertheless, since it is indeed paradoxical to have no change in growth, but at the same time the gut seems to be healthier histomorphologically.

Reviewer 3 Report
Leblois, J. et al. described very interesting research about the impact of a sow's diet rich in wheat bran on the intestinal architecture and immune gene expression in suckling pigs. The manuscript presents correctly designed and performed scientific research with the use of the adequate methods.
Before publication, several elements should be improved:
- In table 1, the fragment of the table showing the "Overall P-values" data is illegible (mainly the second column - it is not known exactly which data refers to which line). Maybe present them in a separate table or change the present one in a more legible way?
- The obtained results of the relative gene expression in ileum and colon mucosa are very interesting, but they need to be presented in a figure/diagram, thanks to which they will be more understandable and easier to interpret. Please add an extra figure.
Also, please correct some editorial mistakes:
- additional paragraph on line 58 (please delete),
- no space between the word and the parenthesis with p value on lines 130, 139, 151 and 153 (please add a space and check this error throughout the paper),
- in the "Fundings" section, please delete the statement "Please add:".
Author Response
Response to Reviewer 3 Comments
Point 1: In table 1, the fragment of the table showing the "Overall P-values" data is illegible (mainly the second column - it is not known exactly which data refers to which line). Maybe present them in a separate table or change the present one in a more legible way?
Response: We have modified the table, and made it more visual.
Point 2: The obtained results of the relative gene expression in ileum and colon mucosa are very interesting, but they need to be presented in a figure/diagram, thanks to which they will be more understandable and easier to interpret. Please add an extra figure.
Response: we have added Supplementary Tables 3 and 4 with all the information on the gene expression data.
Point 3: Also, please correct some editorial mistakes:
additional paragraph on line 58 (please delete)
no space between the word and the parenthesis with p value on lines 130, 139, 151 and 153 (please add a space and check this error throughout the paper), in the "Funding" section, please delete the statement "Please add:"
Response: we have changed all of these editorial mistakes according your suggestion.
Round 2
Reviewer 2 Report
The authors have addressed all the questions